# Developing Hospital Emergency and Disaster Management Index Using TOPSIS Method

**Mohammad Mojtahedi** [1,*] **, Riza Yosia Sunindijo** [1] **, Fatma Lestari** [2] **, Suparni** [3] **and Oktomi Wijaya** [4]

1   School of Built Environment, UNSW Sydney, Sydney, NSW 2052, Australia; r.sunindijo@unsw.edu.au
2   Faculty of Public Health, Universitas Indonesia, Depok 16424, Indonesia; fatma@ui.ac.id
3   Sekolah Tinggi Ilmu Kesehatan Dharma Husada, Bandung 40282, Indonesia; nsuparni@stikesdhb.ac.id
4   Faculty of Public Health, Universitas Ahmad Dahlan, Yogyakarta 55166, Indonesia;
    oktomi.wijaya@ikm.uad.ac.id
*   Correspondence: m.mojtahedi@unsw.edu.au

**Abstract:** Indonesia is a country prone to experiencing natural hazards and disasters, which have frequently damaged public infrastructure, including hospitals. The role of hospitals is crucial to alleviate the impact of disasters. However, there is still a lack of study that analyzes the factors that influence the readiness of hospitals in emergency situations. Filling in this gap, the aim of this paper is to analyze and rank hospitals across West Java and Yogyakarta, Indonesia by the resilience of their emergency management approaches. This research seeks to measure hospital resiliency during emergencies and disasters. Results indicate that the emergency and disaster management coordination, response and disaster recovery planning, communication and information management, logistics and evacuation, human resources, finance, patient care and support services, decontamination and security are key attributes for the decision-making matrix. Based on the Hospital Safety Index tool, this research proposes the Hospital Emergency and Disaster Management (HEDM) index by combining the key attributes and sub-attributes using the Technique for Order of Preference by Similarity to Ideal Solution (TOPSIS) as a multi-attribute decision-making technique. The paper concludes that the anticipated benefits of analyzing the resilience of hospitals by using HEDM is the identification of the most susceptible hospitals based on their levels of readiness and resiliency in areas which are prone to experiencing disasters. This prioritization is important for resource allocation and budget planning.

**Keywords:** hospital emergency and disaster management; hospital resiliency; hospital safety index; Indonesia; TOPSIS

## 1. Introduction

Over the past decades, significant growth in the frequency, scale and intensity of natural hazards including pandemics, wildfires, terrorist attacks, earthquakes, storms and major floods has had devastating impacts on the societies and built environments [1]. The health infrastructures, particularly in developing countries, are vulnerable to the impact of natural hazards [2,3]. Often hospital buildings are damaged by disasters and, as a result, their health service delivery is significantly compromised. Although numerous researchers have paid attention to the critical role of hospitals in society to serve injuries in emergency conditions, less attention has been devoted to the preparedness, recovery, and resilience of hospitals [4].

It is expected that hospitals need to be fully operational during and after disasters. World Health Assembly made a resolution to achieve this aspiration in 1981. Over the years, preventive measures and preparedness for emergencies were established in the health sector globally [5]. Despite considerable progress, many hospitals in disaster-prone areas are still unprepared and, as a result, are not functioning during and after disasters [6].

As a disaster-prone country, Indonesia experiences frequent occurrences of hazards and disasters, and it is located in a hazardous region where 90% of the world's earthquakes occur. In addition, Indonesia is home to 15% of the world's active volcanoes, making the occurrence of volcanic eruptions, earthquakes, and tsunamis regular [7,8]. The country also has thousands of rivers passing through urban areas. Along with heavy rains, illegal houses along riverbanks, and a rubbish-clogged and outdated sewerage system, floods are another serious hazard [7]. Climate change, rapid urbanization, and deforestation can worsen the impacts of disasters in Indonesia, making hospital disaster preparedness a particularly serious and urgent matter.

These disasters have damaged and destroyed hospitals in Indonesia, rendering them nonfunctioning. In 2006, an earthquake closed 17 hospitals in Yogyakarta, while 45 health centers (nearly 40% in the area) were destroyed [9]. Eighty-five hospitals and health facilities were damaged by an earthquake in Padang in 2009 [10]. In 2018, an earthquake and tsunami struck Palu and rendered all hospitals in the city inoperative [11]. The role of hospitals in disaster management is irreplaceable because their ability to deliver health services during these emergencies is a matter of life and death. It is crucial for these health facilities to be safe, accessible, and functioning at an optimum capacity during and after disasters [12]. Therefore, assessing the disaster preparedness of hospitals in Indonesia is crucial for developing resilient cities.

There are many evaluation checklists and tools for assessing hospital disaster preparedness [6]. One such tool is the Hospital Safety Index (HSI), which measures a hospital's operational capacity in disaster and emergency situations. HSI is also intended to help decision-makers identify hospitals which require immediate interventions to enhance their safety and operationality [12]. HSI was introduced in 2018 and now has been used widely in many countries. In Latin America, the index is used in 28 countries and territories. In some European countries, HSI has been integrated into accreditation of hospitals, planning for new hospitals, and hospital improvement programs. The Iranian Government used the index to evaluate 900 hospitals and allocate resources to hospitals that required urgent improvements. The Indonesian standard now requires hospitals to assess their disaster preparedness using the HSI [13].

Previous research shows that there are tools, checklists, and frameworks to assess hospital disaster preparedness and they have been used in various countries. However, there is a lack of clear, quantitative decision-making tools available to measure or benchmark the effectiveness of alternative hospital-emergency and disaster-management strategies. In addition, research on hospital disaster preparedness in Indonesia is rare, despite the country's susceptibility to disasters. This research, therefore, contributes to filling in this gap in existing knowledge. Furthermore, both HSI and TOPSIS technique have been used widely and are proven to be robust. Integrating them in developing HEDM index for Indonesia is a unique contribution of this research.

Building on the efforts of HSI, this research further develops HSI to be a comprehensive tool or index to compare the resilience of emergency disaster management of hospitals. This research, therefore, aims to assess the preparedness of hospitals in managing disasters in emergency situations by proposing Hospital Emergency and Disaster Management (HEDM) index of hospitals in Indonesia. The Technique for Order Preference by Similarity to Ideal Solution (TOPSIS) was selected to develop the HEDM index. TOPSIS is a multiple-criteria decision-making (MCDM) method developed to solve real-world decision problems. It has been used successfully in various application areas, including disaster management, and its application among researchers and practitioners has grown exponentially [14].

## 2. Hospital Disaster Preparedness Tools and Indices

Most of the previous studies have focused on improving surge capacity for enhancing hospital disaster preparedness [15,16]. The American College of Emergency Physicians defined hospital disaster preparedness as the "healthcare system's ability to manage a sudden or rapidly progressive influx of patients within the currently available resources at

a given point in time" [17]. This definition, however, does not cover all aspects of hospital preparedness, such as coordination, response and recovery planning, communications and human resource management, logistic and evacuation.

Much research has been conducted in a different part of the world showing that healthcare systems are not well equipped to handle the impact of disasters, for example, studies in developing countries like Turkey [18], Sri Lanka [19], Serbia [3], India [20], the Kingdom of Saudi Arabia [21], Yemen [22], and also Hong Kong [23]. In fact, a lack of hospital disaster preparedness is becoming more frequent in developed countries as shown by studies in Canada [24] and USA [15]. Although some infrastructure resiliency has been improved by developing and developed countries' governments, inadequate disaster preparedness remains at hospitals [19]. Effective hospital disaster preparedness is an essential element of disaster preparedness in emergency jurisdictions, and agile and effective hospital services can remarkably reduce the mortality rate in the event of a disaster [25].

The need for hospitals to be fully operational in disasters is widely recognized in the literature. Nekoie-Moghadam et al. [6] conducted a systematic review of tools and checklists used for the evaluation of hospital disaster preparedness published from 1990 to 2013. The objective of their research is to prepare a standardized tool to evaluate hospital disaster preparedness. They found 15 tools and suggested 14 hospital disaster preparedness elements. HSI (earlier edition) is one of the tools included in their research, demonstrating the relevance of the tool.

In other research, Fallah-Aliabadi et al. [26] systematically reviewed 32 articles and guidelines published before September 2018 to identify indicators of hospital disaster resilience. HSI is one of the guidelines selected and analyzed in their research. They recommended that the indicators can be categorized into three domains: constructive, infrastructural, and administrative resilience. This categorization is similar to HSI, which focuses on structural safety, nonstructural safety, and emergency and disaster management.

Another systematic review was conducted by Labrague et al. [27]. Different from the previous two reviews that focus on general disaster preparedness of hospitals, this review focuses on a specific group of health service personnel, i.e., nurses and their preparedness for disaster response. Seventeen articles published from 2006 to 2016 were analyzed, and they found that nurses are not sufficiently prepared to respond effectively to disasters. They recommended that future research should identify factors that support disaster preparedness in nurses.

Cole et al. [28] used the Haddon matrix to combine engineering concepts, behavioral sciences, and legal factors into hospital preparedness for earthquakes. This matrix aids users in organizing identified factors that influence and contribute to outcomes of interest, such as hospital preparedness, during and after selected events, such as disasters.

Dell'Era et al. [29] compared Swiss hospital disaster preparedness in 2006 and 2016. A questionnaire was specifically developed in their research to assess hospital preparedness in managing various disaster situations. They found that the rate of hospitals with a disaster plan has increased considerably, but the health care system is still vulnerable to specific threats. There are other studies that focus on assessing hospital disaster preparedness in a specific geographical location. For example, Paganini et al. [30] interviewed Italian emergency physicians to assess their knowledge on basic disaster planning and procedures. They found that the physicians' knowledge base is poor, demonstrating the need for training to ensure that hospital disaster plans are known by all who are responsible for disaster risk reduction and management capacity. Sayed et al. [31] reported their lessons learned in modifying a hospital disaster preparedness plan for mass casualty incidents based on the downtown Beirut bombing. Naser et al. [32] assessed hospital disaster preparedness in South Yemen and found that eight of 10 hospitals had unacceptable levels of preparedness.

Emergency authorities, local governments, hospital managers, urban planners and many other stakeholders have demonstrated the need to evaluate their urban resilience

efforts with robust quantitative indices. This type of indicator or index will enable local decision makers to evaluate the need for actions pertinent to resilience and the value of their investments in these areas [10]. Decision-making indicators have been used widely to improve socioeconomic conditions, including social vulnerability, human development, quality of life and emergency preparedness [33]. Over the past decades, some critical indices have been developed for disaster risk management including Coastal City Flood Vulnerability Index (CCFVI), Hurricane Disaster Risk Index (HDRI), Vulnerability Index (VI), and Disaster Preparedness Index (DPI). In essence, these indices are all instances of a disaster index (DI) and play an essential role in managing disaster risks. NDI also supports planning decisions, resource allocation and public education efforts [33]. NDIs are attractive because they combine technical information in a simple and understandable way that people can easily use. NDIs are also useful to compare vulnerability across communities; to facilitate an efficient allocation of limited resources; and to better understand community preparedness. One of the limitations with these developed tools and techniques is that they do not consider alternative stakeholder strategies relating to their collective mitigation, preparedness, response, and recovery activities.

One of the useful tools for developing indices in decision-making studies has been TOPSIS. A comprehensive study was conducted by [34] to review and compare the TOPSIS with other MCDM tools; their findings showed that in general, TOPSIS has shown increasing recognition of powerful MCDM techniques to support strategic decisions. Many researchers have been utilizing the TOPSIS tool in prioritization of alternatives against atttibutues. For example, [35] used TOPSIS for developin a synthetic index of multicriteria derivation as a useful reference in the decision-making processes relating to restructuring and debt relief operations in real estate credit risk assessment.

Although there are many benefits to using TOPSIS because it is simple and easy to understand, computing is efficient, and it can treat complex problems, this method has also shown some deficiencies in assuming the weights of attributes as equal; however, recently, researchers introduced a novel TOPSIS approach in which the decision maker is not able or does not want to fix exact weights for the decision criteria [36]. Although TOPSIS was developed in 1981 by Hwan [37], TOPSIS as a decision-making tool has been used in many scientific and practical applications, for example in construction risk management [38], supply chain management [39], manufacturing [40], smart cities [41], robotics [42], human resource management [43], sustainability [44], resilience [45], and disaster management [46].

Some decision-making tools and techniques have been used in emergency disaster management such as AHP [47], intelligent decision making [48], fuzzy approach [49], and special decision support system [50]. In most of the previous research on using decision-making tools for managing disaster risks, the results have been robust and reliable, which indicates the practicality of using these tools; although the TOPSIS has reported higher performance in decision-making in comparison with other tools, TOPSIS has not been used frequently in emergency disaster management studies. Therefore, this research proposes a conceptual framework for using TOPSIS for developing a hospital emergency and disaster management index.

## 3. Conceptual Framework for Hospital Emergency and Disaster Management Index

Hospital emergency and disaster management is defined by [12] as the preparedness of the hospital in response to emergencies and disasters. Hospital emergency and disaster management includes coordination, planning, response, recovery, communication in emergency situations during and after a disaster; based on this guideline, the definition also encompasses the logistics, finance and human resource management and how the patients and staff are supported, and finally the evacuation of the affected patients and staff. As described in its guide, HSI is used to assess a hospital's resilience to emergency and disaster situations. HSI can be used to prioritize hospitals that require urgent improvements.

As shown in Figure 1 and aligning with the research aim on assessing the disaster preparedness of the hospitals, the HEDM index consists of 40 items (Module 4 in HSI), which are classified into seven dimensions: (1) coordination of emergency and disaster management activities, (2) hospital emergency and disaster management response and recovery planning, (3) communication and information management, (4), human resources, (5) logistics and finance, (6) patient care and support services, and (7) evacuation, decontamination and security. All the attributes are presented in Table 1. This index evaluates the level of the preparedness of a hospital and its personnel, and of its essential operations to provide health services in response to an emergency or disaster.

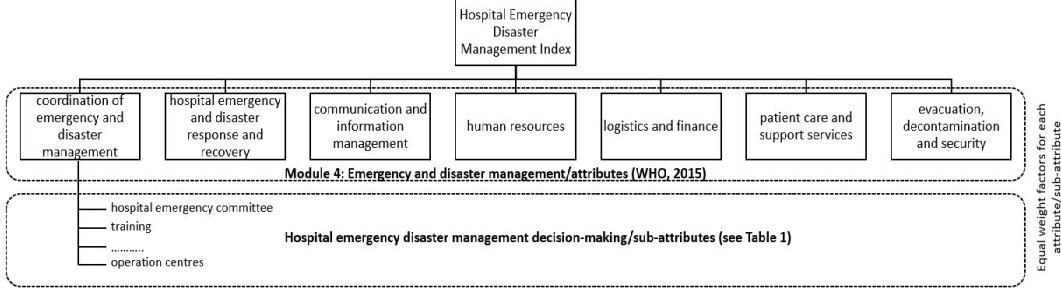

**Figure 1.** Hospital emergency and disaster management indices breakdown.

**Table 1.** Hospital disaster management attributes adopted from HSI [12].

| Code | Attributes |
|---|---|
| **EDM1** | **Coordination of emergency and disaster management activities** |
| EDM1.1 | Hospital Emergency/Disaster Committee |
| EDM1.2 | Committee member responsibilities and training |
| EDM1.3 | Designated emergency and disaster management coordinator |
| EDM1.4 | Preparedness programme for strengthening emergency and disaster response and recovery |
| EDM1.5 | Hospital incident management system |
| EDM1.6 | Emergency Operations Centre (EOC) |
| EDM1.7 | Coordination mechanisms and cooperative arrangements with local emergency management agencies |
| EDM1.8 | Coordination mechanisms and cooperative arrangements with the health-care network |
| **EDM2** | **Hospital emergency and disaster response and recovery planning** |
| EDM2.1 | Hospital emergency or disaster response plan |
| EDM2.2 | Hospital hazard-specific subplans |
| EDM2.3 | Procedures to activate and deactivate plans |
| EDM2.4 | Hospital emergency and disaster response plan exercises, evaluation and corrective actions |
| EDM2.5 | Hospital recovery plan |
| **EDM3** | **Communication and information management** |
| EDM3.1 | Emergency internal and external communication |
| EDM3.2 | External stakeholder directory |
| EDM3.3 | Procedures for communicating with the public and media |
| EDM3.4 | Management of patient information |
| **EDM4** | **Human resources** |
| EDM4.1 | Staff contact list |
| EDM4.2 | Staff availability |
| EDM4.3 | Mobilization and recruitment of personnel during an emergency or disaster |
| EDM4.4 | Duties assigned to personnel for emergency or disaster response and recovery |
| **EDM5** | **Logistics and finance** |
| EDM5.1 | Agreements with local suppliers and vendors for emergencies and disasters |
| EDM5.2 | Transportation during an emergency |
| EDM5.3 | Food and drinking-water during an emergency |
| EDM5.4 | Financial resources for emergencies and disasters |

**Table 1.** *Cont.*

| Code | Attributes |
|------|-----------|
| **EDM6** | **Patient care and support services** |
| EDM6.1 | Continuity of emergency and critical care services |
| EDM6.2 | Continuity of essential clinical support services |
| EDM6.3 | Expansion of usable space for mass casualty incidents |
| EDM6.4 | Triage for major emergencies and disasters |
| EDM6.5 | Triage tags and other logistical supplies for mass casualty incidents |
| EDM6.6 | System for referral, transfer and reception of patients |
| EDM6.7 | Infection surveillance, prevention and control procedures |
| EDM6.8 | Psychosocial services |
| EDM6.9 | Post-mortem procedures in a mass fatality incident |
| **EDM7** | **Evacuation, decontamination and security** |
| EDM7.1 | Evacuation plan |
| EDM7.2 | Decontamination for chemical and radiological hazards |
| EDM7.3 | Personal protection equipment and isolation for infectious diseases and epidemics |
| EDM7.4 | Emergency security procedures |
| EDM7.5 | Computer system network security |

HSI [12] recommends the combined use of structured observation, document review, and interview to assess the items. For instance, an item called 'Hospital Disaster Committee' (coordination of emergency and disaster management activities) can be assessed by interview and document review. The interview is used to confirm whether a committee has been arranged to manage hospital emergency response, while the document review is conducted to review the committee's terms of reference and the list of associates. Another item called 'transportation during an emergency' (logistics and finance) is assessed by using document review and observation. The document review is used to verify whether procedures are in place to ensure availability and access to transport facilities, while the observation is to verify whether such transport facilities are available in the case of emergencies.

HSI [12] also provides guidelines on the scoring process. Each item is scored either low (0 score), average (0.5 score), or high (1 score). When there is more than one evaluator, which is common in the process of collecting data, a consensus is used to finalize the value of each item. Using the 'transportation during an emergency' item as an example, the guidelines use the following descriptions to represent each score level: Low = Transportation system, ambulances are not available; Average = Some vehicles are available, but not enough for a major emergency situations; High = Enough suitable vehicles are available during disasters [13].

Lastly, the ratio for each dimension and module for a hospital can be determined after scoring. Table 2 gives an example of calculating a ratio for logistics and finance. The maximum score for an item is 1 when the item is scored high. Therefore, the maximum score for this dimension is 4 because there are four items representing this dimension. Hospital 1 obtains a score of 2 for this dimension. As such, the ratio of the hospital is 0.500 (hospital score divided by the maximum score).

**Table 2.** Calculating an HEDM guided by HSI [12].

| Dimension | Max Score | Hospital 1 Score | Ratio |
|-----------|-----------|------------------|-------|
| Logistics and finance | 4 | 2 | 0.500 |

Following the HSI guide, below is the meaning of the index score:

- HSI 0–0.35, level C: Urgent intervention measures are needed. The current levels of disaster management measures are not sufficient to protect the lives of patients and hospital staff in disaster events.

- HSI 0.36–0.65, level B: Intervention measures are needed in the short term. Patients, hospital staff, and the hospital's ability to function during and after emergencies and disasters are potentially at risk.
- HSI 0.66–1, Level A: The hospital situation is normal. However, it is recommended to improve emergency, disaster management capacity and safety level in case of emergencies and disasters.

## 4. Research Methods

This research collected data from West Java and Yogyakarta, two Indonesian provinces which experience regular emergency and disaster scenarios. The National Disaster Management Agency classifies West Java as a disaster-prone area. Some serious disasters include a 6.5-magnitude earthquake in 2017, a tsunami in 2018 that killed hundreds of people in 2018, and flash floods and landslides in 2016 that displaced thousands [51]. Likewise, Yogyakarta experienced major disasters previously. In 2006, a 6.3-magnitude earthquake killed 6000 people and destroyed or damaged 600,000 houses [52]. While recovery was nearly complete, Mount Merapi erupted in 2010, killing 300 people. As reported by the World Bank [53], the eruption also displaced 350,000 people. In 2019, Cyclone Savannah damaged infrastructure and houses, caused floods and landslides, and displaced more than 5000 people.

This research used nonprobabilistic purposive sampling because access to data depended on the access granted by local authorities. Three evaluators used the HSI to collect data from 15 hospitals, of which 10 are located in West Java and 5 in Yogyakarta. The three evaluators were a public health officer, an engineer, and a physician. They all have experiences in hospital disaster management.

As stated earlier, for developing Hospital Emergency and Disaster Management (HEDM) Index, multiple-attribute analysis based on TOPSIS was selected to assess the vulnerability of hospitals in Indonesia against emergencies and disasters.

Figure 2 presents the three phases of the TOPSIS technique for its application in Indonesian hospitals. Seven emergency disaster management main categories for developing HSI are proposed in Phase 1, and those seven attributes have been further broken down into 40 sub-attributes as input criteria for TOPSIS. In Phase 3, we consider the expert evaluation on each sub-attribute by applying the TOPSIS method.

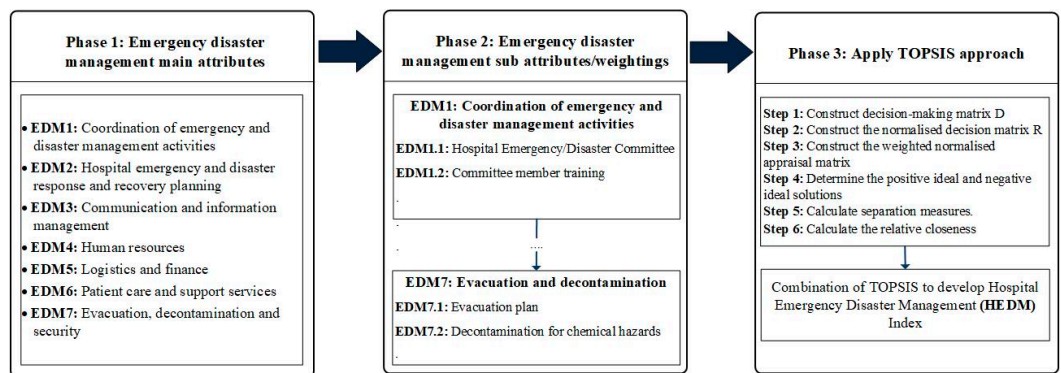

**Figure 2.** Proposed TOPSIS approach for hospital emergency and disaster management index.

Based on the principles of multi-attribute group decision-making (MAGDM), a mathematical optimization model was used to integrate the factors identified in Phases 1 and 2 into an integrated decision-making matrix that characterizes the emergency disaster management approaches of hospitals in Indonesia. MAGDM is an optimization technique which is commonly used to tackle a conflicting condition problem. This approach aims to determine the most desirable approach to reduce risks and to generate the highest level of stakeholder satisfaction. In MAGDM, the alternatives that are associated with com-

mensurate or conflicting attributes are selected or ranked. For this purpose, an MAGDM technique is required to index the various factors.

## 5. Technique for Order of Preference by Similarity to Ideal Solution (TOPSIS)

The TOPSIS procedure proposed by Hwang et al. [37] is as follows:

Step 1: Construct decision-making matrix $D = [d_{ij}]$. $d_{ij}$ indicates the performance rating of $i$th hospital with respect to $j$th emergency disaster management factor.

Step 2: Construct the normalized decision matrix $R = [r_{ij}]$. The vector-normalized value $r_{ij}$ in the decision matrix $R$ can be calculated by Equation (1):

$$r_{ij} = \frac{d_{ij}}{\sqrt{\sum_{j=1}^{n}(d_{ij})^2}}, \qquad i = 1, 2, 3, \ldots, m \tag{1}$$

Step 3: Develop the weighted normalized decision matrix. Each criterion cannot be assigned equal importance because the attributes have multiple interpretations. The weighted normalized matrix is calculated by multiplying the normalized matrix $r_{ij}$ by its associated weight $x_j^*$ to get the result. The weighted normalized value is presented by Equation (2):

$$v_{ij} = x_j^* \times r_{ij}; \qquad i = 1, 2, 3, \ldots, m, \qquad j = 1, 2, 3, \ldots, n \tag{2}$$

$x_j^*$ is the weight of each emergency disaster management factor which is assumed to be equal among all factor techniques where;

$$\sum_{j=1}^{n} x_j^* = 1, \tag{3}$$

Step 4: Calculate the positive ideal and negative ideal solutions. The PIS $(V^+)$ and NIS $(V^-)$ are shown as Equations (4) and (5):

$$PIS \ (V^+) = (v_1^+, v_2^+, \ldots, v_n^+) = \left\{ \left( \max_i v_{ij} \middle| i = 1, 2, \ldots, m \right), \quad j = 1, 2, \ldots, n \right\} \tag{4}$$

$$NIS \ (V^-) = (v_1^-, v_2^-, \ldots, v_n^-) = \left\{ \left( \min_i v_{ij} \middle| i = 1, 2, \ldots, m \right), \quad j = 1, 2, \ldots, n \right\} \tag{5}$$

Step 5: Calculate separation measures. The distance of each alternative from $V^+$ and $V^-$ can be currently calculated using Equations (6) and (7).

$$d_i^+ = \left\{ \sum_{j=1}^{n} \left( v_{ij} - v_j^+ \right)^2 \right\}^{0.5}, \quad i = 1, 2, \ldots, m \tag{6}$$

$$d_i^- = \left\{ \sum_{j=1}^{n} \left( v_{ij} - v_j^- \right)^2 \right\}^{0.5}, \quad i = 1, 2, \ldots, m \tag{7}$$

Step 6: Calculate the relative closeness to the ideal solution which is called $HEDM_i$. This step solves the similarities to an ideal solution by Equation (8):

$$HEDM_i = \frac{d_i^-}{d_i^+ + d_i^-}, \quad i = 1, 2, \ldots, m \tag{8}$$

## 6. Application of TOPSIS to Indonesian Hospital Emergency and Disaster Management

In this paper, a noncompensatory approach is used for the prioritization of hospitals for emergency disaster management, using the TOPSIS, which is a commonly used MAGDM method.

TOPSIS is a powerful technique for analyzing and prioritizing alternatives. It uses the positive ideal solution (PIS) value from the best solution and negative ideal solution (NIS) value from the worst solution. Both PIS and NIS distances to calculate a Net Concordance Dominance (NCD) value are considered using TOPSIS [38]. The idea of the NCD notion stems from the prospect theory which is used to identify the ideal point from which a compromised solution would have the shortest distance from the ideal solution. In this paper, TOPSIS and NCD develop score values for each hospital's HEDM index. Table 3 shows the decision-making matrix $D = [d_{ij}]$. $d_{ij}$ indicates the performance rating of $i$th hospital with respect to $j$th emergency disaster management factor. As explained in Section 2, [12] also provides guidelines on the scoring process. Each item (each hospital) in Table 3 is scored either high, average, or low, in which the value of each level is 1, 0.5, and 0 respectively against emergency and disaster management activities. For example, West Java 1 hospital (WJ1) has shown average performance or engagement related to Hospital Emergency/Disaster Committee (EDM1.1), so a score of 0.5 is allocated to WJ1 for Hospital Emergency/Disaster Committee.

**Table 3.** Decision-making matrix for developing the HEDM index.

|        | WJ1 | WJ2 | WJ3 | WJ4 | WJ5 | WJ6 | WJ7 | WJ8 | WJ9 | WJ10 | Y1 | Y2 | Y3 | Y4 | Y5 |
|--------|-----|-----|-----|-----|-----|-----|-----|-----|-----|------|----|----|----|----|----|
| EDM1.1 | 0.5 | 0.5 | 0.5 | 0 | 0.5 | 0.5 | 0.5 | 1 | 0.5 | 0.5 | 1 | 1 | 1 | 1 | 0.5 |
| EDM1.2 | 0 | 0.5 | 0.5 | 0 | 0.5 | 0 | 0.5 | 0.5 | 0.5 | 0.5 | 0.5 | 0.5 | 0.5 | 0.5 | 0.5 |
| EDM1.3 | 0.5 | 0.5 | 0.5 | 0.5 | 0.5 | 0.5 | 0.5 | 1 | 0.5 | 0.5 | 1 | 0.5 | 1 | 1 | 1 |
| EDM1.4 | 0.5 | 0.5 | 0.5 | 0 | 0 | 0.5 | 0.5 | 0.5 | 0.5 | 0.5 | 0 | 0 | 0 | 0 | 0 |
| EDM1.5 | 0 | 0.5 | 0.5 | 0 | 0.5 | 0.5 | 0.5 | 1 | 0.5 | 0.5 | 0.5 | 0 | 0.5 | 0.5 | 0.5 |
| EDM1.6 | 0 | 0 | 0 | 0 | 0 | 0 | 0 | 1 | 0 | 0 | 0.5 | 1 | 0.5 | 0.5 | 0.5 |
| EDM1.7 | 0.5 | 0.5 | 0.5 | 0 | 1 | 0.5 | 0.5 | 1 | 1 | 1 | 0.5 | 0 | 0.5 | 0 | 0 |
| EDM1.8 | 0.5 | 1 | 0.5 | 0 | 1 | 0.5 | 1 | 1 | 1 | 1 | 0.5 | 0 | 0.5 | 0 | 0.5 |
| EDM2.1 | 0.5 | 0.5 | 0.5 | 0 | 0.5 | 0.5 | 0.5 | 1 | 0.5 | 1 | 0 | 0 | 0.5 | 0 | 0 |
| EDM2.2 | 0.5 | 0.5 | 0.5 | 0 | 1 | 0.5 | 0.5 | 1 | 0.5 | 0.5 | 0 | 0 | 0 | 0 | 0 |
| EDM2.3 | 0.5 | 0.5 | 0.5 | 0 | 1 | 0 | 0 | 1 | 0.5 | 0.5 | 0.5 | 1 | 0.5 | 0.5 | 0.5 |
| EDM2.4 | 0.5 | 0.5 | 0.5 | 0 | 0.5 | 0.5 | 0.5 | 1 | 0.5 | 0.5 | 0.5 | 0.5 | 0.5 | 0.5 | 0.5 |
| EDM2.5 | 0 | 0 | 0.5 | 0 | 0.5 | 0 | 0 | 1 | 0.5 | 0.5 | 0 | 0 | 0 | 0 | 0 |
| EDM3.1 | 0.5 | 0.5 | 0.5 | 0 | 0.5 | 0 | 0.5 | 1 | 0.5 | 0.5 | 0 | 0.5 | 0.5 | 0.5 | 0 |
| EDM3.2 | 0.5 | 0.5 | 0.5 | 0.5 | 0.5 | 0.5 | 0.5 | 1 | 0.5 | 0.5 | 0 | 1 | 0 | 0 | 0.5 |
| EDM3.3 | 0.5 | 0.5 | 0.5 | 0 | 0 | 0.5 | 0.5 | 1 | 0.5 | 0.5 | 0.5 | 0.5 | 0.5 | 0 | 0 |
| EDM3.4 | 0.5 | 0.5 | 0.5 | 0 | 0.5 | 0.5 | 0.5 | 1 | 0.5 | 0.5 | 0.5 | 0.5 | 0 | 0.5 | 0 |
| EDM4.1 | 0.5 | 0.5 | 1 | 0.5 | 0 | 0.5 | 0.5 | 0.5 | 0.5 | 0.5 | 0 | 0.5 | 0.5 | 0 | 1 |
| EDM4.2 | 0.5 | 0.5 | 0.5 | 0.5 | 0.5 | 0.5 | 0.5 | 1 | 0.5 | 0.5 | 0.5 | 0.5 | 0.5 | 0.5 | 0.5 |
| EDM4.3 | 0.5 | 0.5 | 0 | 0 | 0.5 | 0.5 | 0.5 | 1 | 0.5 | 0.5 | 0.5 | 0 | 0 | 0 | 0 |
| EDM4.4 | 0 | 0 | 0 | 0 | 0.5 | 0.5 | 0.5 | 0.5 | 0.5 | 0.5 | 0.5 | 0 | 0.5 | 0 | 0 |
| EDM4.5 | 0 | 0 | 0 | 0 | 0 | 0 | 0 | 1 | 0 | 0 | 0.5 | 0 | 0.5 | 0 | 0 |
| EDM5.1 | 0.5 | 0.5 | 0.5 | 0.5 | 0.5 | 0.5 | 0.5 | 1 | 0.5 | 0.5 | 0.5 | 0 | 0 | 0.5 | 0 |
| EDM5.2 | 0 | 0.5 | 0.5 | 0.5 | 1 | 0 | 0.5 | 1 | 0.5 | 1 | 1 | 1 | 0.5 | 1 | 1 |
| EDM5.3 | 0.5 | 0.5 | 0.5 | 0.5 | 1 | 0.5 | 0.5 | 1 | 1 | 1 | 0 | 0 | 0 | 0 | 0 |

**Table 3.** *Cont.*

| | WJ1 | WJ2 | WJ3 | WJ4 | WJ5 | WJ6 | WJ7 | WJ8 | WJ9 | WJ10 | Y1 | Y2 | Y3 | Y4 | Y5 |
|---|---|---|---|---|---|---|---|---|---|---|---|---|---|---|---|
| EDM5.4 | 0.5 | 0.5 | 0.5 | 0 | 1 | 0.5 | 0.5 | 1 | 1 | 1 | 0 | 0 | 0 | 0 | 0 |
| EDM6.1 | 0.5 | 0.5 | 0.5 | 0 | 0 | 0.5 | 0.5 | 0.5 | 0.5 | 0.5 | 0 | 0 | 0 | 0 | 0 |
| EDM6.2 | 0 | 0 | 0.5 | 0 | 0.5 | 0 | 0.5 | 0.5 | 0.5 | 0.5 | 0 | 0 | 0 | 0 | 0 |
| EDM6.3 | 0 | 0 | 0.5 | 0 | 0.5 | 0 | 0.5 | 0.5 | 0.5 | 0.5 | 0.5 | 0.5 | 0.5 | 0.5 | 0 |
| EDM6.4 | 1 | 0.5 | 0.5 | 0.5 | 0.5 | 1 | 1 | 1 | 0.5 | 0.5 | 1 | 0.5 | 0.5 | 0.5 | 1 |
| EDM6.5 | 0.5 | 0.5 | 0.5 | 0.5 | 0.5 | 0.5 | 0.5 | 1 | 0.5 | 0.5 | 0.5 | 0.5 | 0 | 0.5 | 0.5 |
| EDM6.6 | 0.5 | 0.5 | 0.5 | 0 | 0.5 | 0.5 | 0.5 | 1 | 0.5 | 0.5 | 0.5 | 0 | 0.5 | 0 | 0.5 |
| EDM6.7 | 0.5 | 0.5 | 0.5 | 0.5 | 1 | 0.5 | 0.5 | 1 | 0.5 | 0.5 | 0 | 0 | 0 | 0 | 0 |
| EDM6.8 | 0 | 0 | 0 | 0 | 0 | 0 | 0 | 0 | 0 | 0 | 0 | 0 | 0 | 0 | 0.5 |
| EDM6.9 | 0 | 0 | 0 | 0.5 | 0.5 | 0 | 0 | 0.5 | 0.5 | 0.5 | 0.5 | 0.5 | 0 | 0.5 | 0 |
| EDM7.1 | 0.5 | 0.5 | 0.5 | 0 | 0.5 | 0.5 | 0.5 | 1 | 0.5 | 1 | 0.5 | 0.5 | 0.5 | 0 | 0.5 |
| EDM7.2 | 0.5 | 0.5 | 0.5 | 0.5 | 1 | 0.5 | 0.5 | 1 | 1 | 1 | 0 | 0 | 0.5 | 0 | 0 |
| EDM7.3 | 0.5 | 0.5 | 0.5 | 0.5 | 1 | 0.5 | 0.5 | 1 | 1 | 1 | 0.5 | 0.5 | 0.5 | 0.5 | 0.5 |
| EDM7.4 | 0.5 | 0.5 | 0.5 | 0 | 0.5 | 0.5 | 0.5 | 1 | 1 | 0.5 | 0.5 | 0.5 | 0 | 0.5 | 0.5 |
| EDM7.5 | 0.5 | 0.5 | 0.5 | 0 | 0.5 | 0.5 | 0.5 | 1 | 0.5 | 0.5 | 0.5 | 0 | 0 | 0.5 | 0 |

The TOPSIS procedure and steps explained in Section 4 have been applied for Table 3 (decision-making matrix), and the results of TOPSIS have been provided in Table 4 which presents the respective HEDM value obtained from the TOPSIS procedure. The table shows that West Java 8 (HEDM = 0.734), West Java 10 (HEDM = 0.507) and West Java 9 (HEDM = 0.494) are the most resilient hospitals against disasters and emergency situations.

**Table 4.** HEDM index of case study hospitals.

| Hospital | PIS | NIS | HEDM Index | Rank |
|---|---|---|---|---|
| West Java 1 | 0.0555 | 0.0316 | 0.363 | 11 |
| West Java 2 | 0.0534 | 0.0340 | 0.389 | 7 |
| West Java 3 | 0.0509 | 0.0373 | 0.423 | 5 |
| West Java 4 | 0.0698 | 0.0184 | 0.209 | 15 |
| West Java 5 | 0.0492 | 0.0455 | 0.480 | 4 |
| West Java 6 | 0.0557 | 0.0323 | 0.367 | 10 |
| West Java 7 | 0.0517 | 0.0373 | 0.419 | 6 |
| West Java 8 | 0.0257 | 0.0707 | 0.734 | 1 |
| West Java 9 | 0.0461 | 0.0449 | 0.494 | 3 |
| West Java 10 | 0.0454 | 0.0468 | 0.507 | 2 |
| Yogya 1 | 0.0567 | 0.0339 | 0.374 | 9 |
| Yogya 2 | 0.0632 | 0.0335 | 0.347 | 12 |
| Yogya 3 | 0.0603 | 0.0299 | 0.331 | 13 |
| Yogya 4 | 0.0644 | 0.0279 | 0.302 | 14 |
| Yogya 5 | 0.0605 | 0.0362 | 0.375 | 8 |

Figure 3 represents HEDM for the case study hospitals, and it shows the variance of HEDM from the most susceptible hospital (West Java 4) to the most resilient hospital (West Java 8). Figure 2 is also helpful for classifying the hospitals into different categories based on their resiliency against natural hazards and emergency situations.

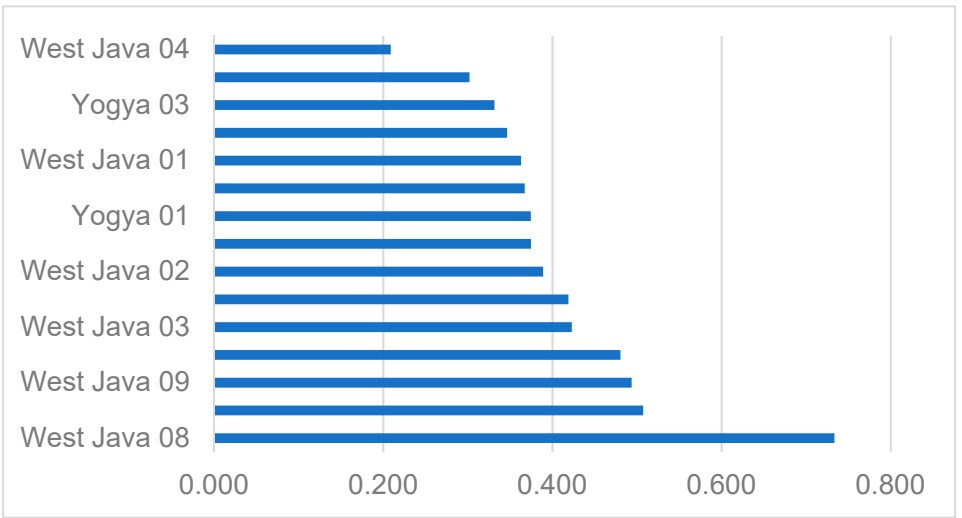

**Figure 3.** HEDM index variability.

## 7. Discussion

Based on the results, there are several findings worth discussing. First, using the HEDM index, ten hospitals in West Java and five hospitals in Yogyakarta were assessed. The average HEDM index of these hospitals is 0.408 out of 1.000 (lower end of B rating according to HSI), which indicates that hospitals in Indonesia are not resilient enough. As shown in the HSI guide, this score means that proactive measures are required in the short-term because the hospitals are not resilient to the impact of disasters. It is particularly concerning to see that the HEDM indices of hospitals in Yogyakarta are relatively low with an average index of 0.346. According to HSI, this score corresponds to a rating of C, which means urgent intervention measures are needed. This rating indicates that the safety, emergency and disaster management approaches are not enough to minimize the risk of disasters and save the lives of patients and hospital staff in emergencies or disasters. It seems that after major disaster events, hospitals in Yogyakarta are still not adequately prepared for disasters.

Second, there are benefits for using the HEDM index particularly for how emergency managers, disaster practitioners, and researchers would be able to use this novel index for better managing hospitals in disasters. For example, the proposed TOPSIS model for developing the HEDM index in this paper allows individual local hospitals to determine and input their own data, measure and compare their own performance and analyze the most effective allocation of resources specific to their situation in emergencies. The benefit of TOPSIS in this context is that in addition to an overall HEDM index, the performance of each hospital can be benchmarked in more specific disaster risk terms and monitored over time. Evaluating indices would enable hospitals and other key stakeholders to allocate hospital resources and initiate resourcing strategies effectively. Maintenance regimes, state-wide funding priorities, insurance premiums, disaster management strategies, urban planning, and/or evacuation planning could then be utilized by hospital managers.

Third, this research has successfully integrated HEDM and TOPSIS. TOPSIS provides a more realistic form of modeling for MAGDM because it allows for trade-offs between attributes. The calculation is uncomplicated, replicable, and can be easily adjusted to reflect any changes in the values of factors. The HEDM index developed from this integration identifies the particular series of factors that best represent the proactive and reactive hospital activities necessary to minimize disaster risk. In fact, the index can be further classified as mitigation, preparedness, response and/or recovery factors, providing an individual index for each emergency and disaster management attribute in hospitals.

Fourth, by employing the method used in this research, Indonesian hospitals can improve their preparedness against disasters and emergencies by the following:

- Identifying the most susceptible hospitals which are located in disaster-prone areas.
- Evaluating the disaster and emergency preparedness of hospitals in vulnerable areas. The results can be used to prioritize the allocation of resources and budgeting.
- Implementing disaster risk management proactive measures as needed. Prioritization is necessary for optimum budget allocation.
- Introducing the appropriate solutions for specific local governments to manage disaster risk that is more specific to the related hospitals and to improve the resilience of the hospitals.

Fifth, Although flood is used as the disaster type in this research project, the proposed HEDM has the potential to be used for decisions for other types of disasters such as earthquake, volcano, fire, tsunami, etc.

## 8. Conclusions

This research has evaluated hospital disaster preparedness in Indonesia and found that Indonesian hospitals are still not adequately prepared despite the fact that the country is notorious for its proneness to disasters. By developing the HEDM index, this research contributes by providing a practical tool to develop hospitals that are resilient and prepared in the case of emergencies and disasters. In this case, a hospital emergency or disaster response plan is an effective strategy for enhancing the disaster preparedness of hospitals in Indonesia, specifically, and in developing countries, more broadly.

The HEDM index also can be used to benchmark hospitals in terms of the key factors for developing disaster-ready hospitals. The values generated from TOPSIS can then be used by the governments and other agencies to prioritize funds and resources to hospitals that urgently require interventions. Essentially, this research provides a quantitative tool which informs decision makers as they set improvement goals over time. From a big-picture perspective, this research incorporates disaster preparedness of hospitals into resilient urban planning and development as a way to support the Sendai Framework for Disaster Risk Reduction and the United Nations Sustainable Development Goals 3.d.

One of the novelties of this research is to develop the HSI to be a comprehensive tool by assessing the preparedness of hospitals in managing disasters in emergency situations by proposing a Hospital Emergency and Disaster Management (HEDM) index of hospitals in Indonesia. The HSI's focus is mainly on safety of hospitals against disasters; however, the proposed HEDM index in this research is a comprehensive tool which combines all the relevant disaster preparedness activities proposed by (World Health Organization, 2015) into an integrated index.

The benefits of using an HEDM index are listed in the Discussion section particularly for how emergency managers, disaster practitioners, and researchers would be able to use this novel index for better managing hospitals in disasters.

Although flood is used as the disaster type in this research project, the proposed HEDM has the potential to be used for decisions for other types of disasters such as earthquake, volcano, fire, tsunami, etc.

**Author Contributions:** Conceptualization, R.Y.S.; methodology, F.L., O.W. & S.; data curation, F.L., S. & O.W.; formal data analysis, M.M.; writing, M.M. & R.Y.S.; writing—review and editing, M.M. & R.Y.S. All authors have read and agreed to the published version of the manuscript.

**Funding:** This research received no external funding.

**Institutional Review Board Statement:** The study was conducted according to the guidelines of the Declaration of Helsinki, and approved by the Ethics Committee of UNSW Sydney (HC17766, 15 September 2017).

**Informed Consent Statement:** Informed consent was obtained from all subjects involved in the study.

**Data Availability Statement:** Not applicable.

**Conflicts of Interest:** The authors declare no conflict of interest.

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
