# Peer review of "Developing Hospital Emergency and Disaster Management Index Using TOPSIS Method"

_sustainability, doi:10.3390/su13095213_

Round 1

Reviewer 1 Report

Aim of the work is to develop and hospital emergency disaster management index using TOPSIS method. The topic addressed, specially in the referenced territorial context, is of high relevance. The sequence of the contents is well structured and the analysis exposed are appropriately described. However, some improvements can be suggested in order to publish the work:

  • Hospital disaster preparedness studies
    • the sub-paragraph's contents seem to be more adequate to the next one because in this is often highlighted the presence (or not) of the HSI index in the study. This is obviously an analysis that have to be done, but if the title remains the same, I suggest to include TOPSIS applications into similar context (ranging from a general index approach to similar applications) in order to provide an exhaustive literature review both on the topic and the techniques. Some studies on the TOPSIS similar approach could be:
      • Locurcio, M., Tajani, F., Morano, P., & Anelli, D. (2021). A Multi-criteria Decision Analysis for the Assessment of the Real Estate Credit Risks. In Appraisal and Valuation (pp. 327-337). Springer, Cham.
      • Zyoud, S. H., & Fuchs-Hanusch, D. (2017). A bibliometric-based survey on AHP and TOPSIS techniques. Expert systems with applications78, 158-181.
      • Liern, V., & Pérez-Gladish, B. (2020). Multiple criteria ranking method based on functional proximity index: un-weighted TOPSIS. Annals of Operations Research, 1-23.
  • Hospital disaster preparedness tools and indices 
    • this subparagraph should be included in the previous one in order to have only one section focused on tools and studies where are highlighted two things: i) the analysis carried out on the same similar topic and the main outputs; ii) how the same techniques adopted could provide a support tools for the same topic and their efficiency (briefly) also in other research fields.
  • Health care system of Indonesia
    • this must be improved or included into the Introduction. If the Authors decide to improve it, must be listed the existing efforts or systems provided by the legislation in order to manage the disaster occurrence. 
  • Research methods
    • The TOPSIS therefore is based on the methodological general steps provided for the HSI index? (Phase 1 and Criteria of Phase 2). 
    • Only three people were interviewed? where are the descriptive statistics of their profiles? (e.g. age, work experience, occupation, etc.)
  • Technique for Order of Preference by Similarity to Ideal Solution (TOPSIS)
    • Please check the algebraic formulas formatting.
  • Other suggestions
    • The calibration process of the outputs is missing. Maybe also a comparison with similar studies which have applied the HSI index should be inserted in order to prove the consistency of the outputs.

Author Response

First of all, I would like to express my sincere gratitude to the reviewer’s insights for providing us rewarding and constructive feedback. We reviewed the manuscript meticulously to address all the reviewer’s comments to make sure additional information, review, and more references are added in the revised manuscript. Responses to the reviewer’s specific comments are attached.

Reviewer 2 Report

The article "Developing hospital emergency disaster management index using TOPSIS method" adopts Health Safety Index (HSI) 's module 4 indicators and employs the TOPSIS method to rank and analyze the risk of hospital and their preparedness to emergency and disasters in Java and Jogjakarta provinces of Indonesia. The authors used various sets of data as suggested in the original method and constructed an index' hospital emergency disaster management (HEDM)'. Although the authors claim a new index, it is not novel in its work and lacks theoretical justification to construct. Below are my comments:

  • The article nicely summarizes the hospital safety index background and uses one of the modules of the index in variable selection. However, the manuscript lacks to justify why another index is required by using the same set of variables. It is unclear what additional information and knowledge this work would produce for emergency managers, disaster practitioners, and researchers. The manuscript is also silent on how various types of disasters are dealt within this framework as the impact of the flood is entirely different from the impact of earthquakes and volcanoes.

  • Another interesting stuff is the use of the term 'emergency disaster management.' It is a new term used by authors in the manuscript, which requires its definition too. In the literature, 'emergency management,' 'disaster management,' and 'emergency and disaster management' are common but not the 'emergency disaster management.'

  • The manuscript provides details about the TOPSIS method with equations in section 5 of the article. However, it does not explain why TOPSIS is chosen in this analysis as the article highlights TOPSIS in the title, abstract, and keywords of the paper. There are several multi-criteria decision-making tools used in the literature for such ranking analysis.

  • The paper uses the term 'natural disaster'. Only hazards are natural in origin, and depending on various social, economic, and political characteristics of a society, it is transformed into disasters. Please do not use natural to denote disasters.

Author Response

First of all, many thanks for the rewarding and constructive feedback provided by the reviewer. We reviewed the manuscript meticulously to address the entire reviewer’s feedback. We added a new section to highlight the missing parts in the literature review in this study.  Additions and corrections have been highlighted with yellow colour in the revised manuscript. Finally, responses to the reviewer’s specific comments have been provided attached file:

Reviewer 3 Report

Determining the resilience of a community and especially hospitals are of high interest. 

The paper presents in a logical order the findings. Methodology and data is detailed and well-presented and backed up with proper literature review. However it would be great to be more detailed about the results. How the results in table 4 are derivedafter table 3 (it is named 1, please change)? Table 3 represents scores of a specific hospital?

Table 4 should be explained more thoroughly. How PIS, NIS affects HEMD Index? What do these scores represent, how can the authors explain the numbers through an example? Have the authors used the method to already damaged and recovered hospitals? Did they encounter any change in the preparedness of such hospitals? From managements point of view and the condition of such hospitals?

Have the authors compared the vulnerability of the building to emergency preparedness so far there has been an engineer among evaluators? By vulnerability I mean an attribute of the building: vulnerability of the hospital to different hazards?

As a conclusion I would suggest to publish the paper with revisions (adding explanation to the results, to table 3 and 4).

Author Response

First of all, many thanks for the rewarding and constructive feedback provided by the reviewer. We reviewed the manuscript meticulously to address the entire reviewer’s feedback. We added a new section to highlight the missing parts in the literature review in this study.  Additions and corrections have been highlighted with yellow colour in the revised manuscript. Finally, responses to the reviewer’s specific comments have been provided in more details in attached file.
